# Methodical Design of Viral Vaccines Based on Avant-Garde Nanocarriers: A Multi-Domain Narrative Review

**DOI:** 10.3390/biomedicines9050520

**Published:** 2021-05-06

**Authors:** Ehsan Raoufi, Bahar Bahramimeimandi, M. Salehi-Shadkami, Patcharida Chaosri, M. R. Mozafari

**Affiliations:** 1Department of Medical Biotechnology, Faculty of Allied Medicine, Iran University of Medical Sciences, Tehran 1449614535, Iran; ehsan.raoufi@gmail.com (E.R.); bahar_bahrami_74@yahoo.com (B.B.); 2Student Research Committee, School of Medicine, Iran University of Medical Sciences, Tehran 1449614535, Iran; mohammad.salehi33@gmail.com; 3Supreme NanoBiotics Co. Ltd. and Supreme Pharmatech Co. Ltd., 399/90-95 Moo 13 Kingkaew Rd. Soi 25/1, T. Rachateva, A. Bangplee, Samutprakan 10540, Thailand; patcharida.chao@gmail.com; 4Australasian Nanoscience and Nanotechnology Initiative (ANNI), Monash University LPO, Clayton, VIC 3168, Australia

**Keywords:** Corona, COVID-19, drug delivery, lipid particles, liposomes, nanoliposomes, SARS-CoV-2, theranostics, tocosome, virus

## Abstract

The current health crisis caused by coronavirus 2019 (COVID-19) and associated pathogens emphasize the urgent need for vaccine systems that can generate protective and long-lasting immune responses. Vaccination, employing peptides, nucleic acids, and other molecules, or using pathogen-based strategies, in fact, is one of the most potent approaches in the management of viral diseases. However, the vaccine candidate requires protection from degradation and precise delivery to the target cells. This can be achieved by employing different types of drug and vaccine delivery strategies, among which, nanotechnology-based systems seem to be more promising. This entry aims to provide insight into major aspects of vaccine design and formulation to address different diseases, including the recent outbreak of SARS-CoV-2. Special emphasis of this review is on the technical and practical aspects of vaccine construction and theranostic approaches to precisely target and localize the active compounds.

## 1. Introduction

Viruses are microscopic infectious agents, which replicate only inside the living cells of host organisms. They infect all types of life forms from microorganisms (including archaea and bacteria) to plants, animals, and humans [1]. There are mainly four groups of viruses, namely: (i) bacteriophages (viruses that infect bacterial cells); (ii) plant viruses (e.g., TMV: tobacco mosaic virus); (iii) animal viruses (e.g., DHV: duck hepatitis virus; SIV: simian immunodeficiency virus); and (iv) human viruses (e.g., coronaviruses) (see Figure 1). Bacterial viruses, known as bacteriophages, are one class of viruses that are identified as the most widespread biological organisms on the earth [2,3]. They bind to bacterial cells and perfuse the viral genome into the cell via lytic or lysogenic cycles [4,5]. The genetic material of these types of viruses consists of either DNA or RNA molecules [6].

Plant viruses, on the other hand, are parasitic organisms that use plant cells as their hosts. They rely on the host cellular resources and other factors essential for their replication cycle and movement [7]. Plant viruses utilize two types of movement, known as local (cell-to-cell movement in the infected leaves) and systemic (long-distance mobility through the plant vascular system). The genome of these viruses is made of ssDNA, dsRNA, positive-single-strand RNA, or negative-single-strand RNA [8,9].

Animal viruses are another group of viruses that cause infection and disease in animal cells. They need to identify a certain host cellular receptor for their entrance into the cell and the initiation of their life cycle [10]. The genome of animal viruses can be made of either DNA or RNA. Certain subgroups of animal viruses are classified as reverse transcribing viruses (also known as RT viruses), examples of which are simian immunodeficiency virus (SIV) and human immunodeficiency virus (HIV) [11].

Human viruses, on the other hand, are defined as specific pathogens that have to confiscate some of the host cellular pathways to survive [12]. These viruses can either cause disease or could be asymptomatic [13,14]. Human viruses manifest a broad range of various structures [15]. Same as other groups of viruses mentioned above, Human viruses contain either RNA or DNA as their genome. Their nucleic acid material can be single- or double-stranded molecule. Human viruses are classified into several subgroups. Some of the human viruses, such as the herpes virus, might stay hidden in the body and be activated under specific circumstances of stress and immune system compromise [16]. Hepatitis B and C viruses are examples of another type of human pathogens that maintain constant existence and virulency in the body, and people are permanent carriers of these pathogens [17,18] (Figure 1).

In the recent years we are facing novel viral pathogens that are defined as new etiological agents [19]. Emergence of these viral infections has notably affected human health and has caused concerns for the health care systems around the world [20]. Vaccination is one of the most potent approaches in the management of viral diseases. In many cases, the vaccine candidate requires protection from degradation and precise delivery to the target cells. This can be achieved by employing different types of drug and vaccine delivery strategies, among which, nanotechnology-based systems seem to be promising alternatives to the conventional vaccinations. The present narrative review aims to provide insight into major aspects of vaccine design and formulation to address different diseases, including the recent outbreak of SARS-CoV-2. This name was chosen based on the fact that the virus is genetically related to the coronavirus responsible for the SARS outbreak of 2003 and the Middle East Respiratory Syndrome (MERS) outbreak of 2012. On 11 February 2020, the International Committee on Taxonomy of Viruses (ICTV) announced “severe acute respiratory syndrome coronavirus 2” (SARS-CoV-2) as the official scientific name of this new pathogen [21]. Special emphasis of this entry is on the technical and practical aspects of vaccine construction and theranostic approaches to precisely localize the payload.

## 2. Vaccination Platforms

Vaccines are medical tools that cause pathogen-specific immune responses in the body [22]. They induce the immune system to produce efficient antibodies that can attach to the antigens strongly and prevent them from infecting cells. In the recent outbreak of SARS-CoV-2, we are facing an urgent need for an efficient vaccine to minimize the threats of the virus on the human health and global economy as soon as possible [23]. Table 1 lists different types of vaccine protocols currently available to tackle viral diseases. Some of different available vaccine protocols are explained in the following sections.

### 2.1. Nucleic Acid-Based Strategies

The nucleic acid-based vaccines are generally selected from either an RNA molecule (e.g., mRNA) or a plasmid or a double-stranded DNA that encode antigens. These antigens provoke humoral and/or cell-mediated immune responses in the body [24,25]. The DNA/RNA -based vaccines are considered flexible, as they allow manipulating the antigens easily. The synthetic DNA/RNA stimulates the protein production, similar to the infection situation, in the target cells. The synthesized protein will be placed in the cytosolic plasma membrane (e.g., endoplasmic reticulum) and protein modification processes can precisely take place [26]. The important point towards this end is that DNA (or RNA) needs a delivery system to achieve its aims. In the context of nucleic acid-based COVID-19 vaccine formulation, liposomes and lipid nanoparticles (LNP), together with certain electrophysiological approaches (e.g., electroporation), in certain cases, appear to be among the most promising systems to deliver the nucleic acid construct to the target cell [27,28,29,30].

### 2.2. Peptide-Based Vaccines

Peptide-based vaccines have been designed against a variety of infectious and tumorigenic diseases [31]. Their design is based on the prediction and evaluation of specific epitopes of the B and T cells [32]. Epitopes are defined as antigenic regions, which are recognized by the immune cells and hold the immunogenic property for these cells [33]. To conduct an accurate cellular immune response in the body, antigenic segments should bind to a certain class of molecules known as Major Histocompatibility Complex (MHC). The antigenic fragments are processed by the MHC molecules, which will eventually present them to T cells. These peptidic epitopes should be linear in order to be able to bind MHC molecules. B cells possess receptors and are responsible for producing antibodies in the immune system [34]. Immunogenic carrier proteins, nanoparticle delivery systems, and potent adjuvants could be combined in the formulation in order to boost the immunogenicity of peptide-based vaccines. Several peptide-based vaccines (such as multi-epitope vaccines) against SARS-CoV-2 were formulated by interaction with T cells (CD4^+^ and CD8^+^) and B cells along with an adjuvant and assembling multi-epitope by using EAAAK linkers [35,36,37].

### 2.3. Pathogen-Based Strategies

#### 2.3.1. Live Attenuated Viral Vaccines

Attenuated vaccines represent a weakened form of various pathogens, e.g., bacteria and viruses (including the yellow fever virus strain 17D) that lead to several different diseases. They induce potent and long-lasting cellular and humoral immunity in the body similar to the natural response, and are considered to be the oldest and the most efficient mode of vaccination [38,39,40]. Different approaches have been used for the production of the attenuated vaccines, such as employment of related non-human viruses (e.g., rotavirus, smallpox), administration at different body sites (adenovirus), and laboratory adaptation (polio and others) [41].

#### 2.3.2. Inactivated Viral Vaccines

Whole virus vaccines utilize the complete particle of the virus that has been deactivated by processes employing certain chemical compounds, radiation, or heat. The immune responses of these type of vaccines qualitatively differ from those vaccines that are processed through intracellular pathways. In addition, inactivated viral vaccines often need extensive extra trials to confirm their safety because some studies have indicated increased infectivity following immunization [42,43,44].

#### 2.3.3. Viral Subunit Vaccines

Subunit vaccines are specific antigenic segments (such as synthetic peptide or protein) of a pathogen, that are used to induce an immune response and provoke acquired immunity against the pathogen from which they are derived [45,46]. They possess high safety profiles, specifically targeting well-defined neutralizing epitopes [45,47]. There is no risk of incomplete inactivation, regain of virulence of the attenuated virus, or unfavorable host responses to viral subunit vectors [46,48]. Since the prevalence of SARS in 2002, subunit vaccines based on this virus have been studied and tested widely, demonstrating adequate efficacy and protection against SARS-CoV infections in several animal models [49,50,51,52].

### 2.4. Adjuvants

Adjuvants are pharmacological or immunological agents that are being used to improve immune responses. In other words, they are capable of increasing the biological half-life of vaccines and antigen uptake by the antigen-presenting cells (APCs). Currently, many adjuvants with different mechanisms of action are being utilized in vaccine constructs. Examples of adjuvants include mineral salts, saponins, cytokines, microbial components, microparticles, emulsions, and certain liposomes and nanoliposomes. They induce the production of immune regulatory cytokines, activate inflammations, local inflammation, cellular recruitment, and induce more rapid, broader, and stronger immune responses that are essential for good immunogenicity [53,54]. Nanoparticles and nanovesicles, due to their intrinsic adjuvanticity (by activating complement system, inducing autophagy and activation of inflammasome) are also considered as vaccine adjuvants nanosystems [55,56,57,58].

## 3. Nanotechnology in Drug and Vaccine Delivery

Nanotechnology techniques provide substantial advantages in the formulation of novel vaccines and have significantly enhanced their effectiveness [57]. A broad range of active moieties can be encapsulated and delivered by nanovesicles and nanocarriers. These nanosystems possess more surface area to volume ratio when compared to micrometric drug carriers and, as such, they can attach to/interact with the target cells more efficiently. Nanotechnology-based vaccines can be formulated employing nucleic acids, purified subunit proteins or polysaccharides, synthetic peptides, or recombinant proteins. These classes of vaccines may stand in need of adjuvants to become sufficiently immunogenic, as explained above. As explained in the previous section, nanovesicles are able to function as adjuvants for viral vaccines; therefore, they are regarded as nano-adjuvants (NAs) [55,58]. They can also increase immunogenicity, cellular uptake, and stability of vaccines by encapsulation or absorption of the vaccine antigen or DNA in a proper formulation. Moreover, it is postulated that various designs of nanoparticles are able to modulate the systemic release of a vaccine and its bio-distribution [58]. The recent SARS-CoV-2 pandemic has seriously put human health in danger and has resulted in an extensive magnitude of mortality and morbidity all around the world [59]. In the context of this outbreak, emerging nanotechnologies, including some mRNA-based vaccines carried by lipidic nanovesicles and viral vectors, have reached Phase II and III clinical trials and some of such formulations have even been approved for human use so far [60,61,62,63].

An important feature of nanotechnology applications in drug delivery is the possibility of elimination of adverse drug effects through a theranostic approach. Theranostic drug delivery employs the combined features of diagnosis and therapy, and hence, is extremely beneficial in the development of targeted drug delivery systems. Theranostic approach is utilized to design and formulate biocompatible and efficient protocols for stimuli-responsive, controlled vaccine, or drug delivery for various applications [61,62]. The success of theranostics depends on multidisciplinary collaborations, combined with our understanding of host responses to customized therapeutic or preventive agents and formulating individualized therapies.

Associating the right therapeutic moiety to the right nanocarrier or nanovesicle, aimed for a specific disease condition, is essential for the clinical success of nanomedicine against the SARS-CoV-2 and other viral infections. Nanotechnology and nanomedicine enable reformulating different drug candidates to improve their “therapeutic index”, mainly by addressing the limitations associated with the drug molecule and alleviating the conventional toxicity or side effects [59,60,61]. The following sections bring more insight into the micro- and nano-carrier systems for vaccine delivery.

## 4. Vaccine Delivery Techniques

A perfect vaccine delivery technique requires encompassing a mixture of high loading capacity, prolonged half-life, no premature leakage or release of the active compound, no intervention with the consistency of the therapeutic molecule, and a simple, scalable, and reproducible manufacturing procedure. Acceptable bio-compatibility, bio-degradability, and safety aspects of the formulation, in addition to a clear understanding of the mechanism of action of the vaccine delivery system, are also desirable factors. The current COVID-19 pandemic has increased the necessity of constructing safe and efficacious vaccine delivery systems [61]. Nanotechnology and nanomedicine provide substantial advantages for the formulation of vaccine protocols and enable either the targeted delivery or co-delivery of the therapeutic molecules and reinforce the delivery of high-value and high-tech bio-therapeutic agents, such as proteins or genetic material [62]. Some of the main vaccine encapsulation and targeting techniques are presented in the following sub-sections.

### 4.1. Polymeric Carriers

Polymer science possesses an efficient role in producing nanomaterials in a simple, cost-beneficial, and scrutinized way [64]. The design of polymeric therapeutic delivery technologies and vaccine carriers (especially biodegradable and biocompatible constructs) are considered as an essential part of the design and formulation of vaccines [65]. Certain features, such as size, charge, structure, hydrophobicity, aggregation, and natural or synthetic composition, can influence the immunogenicity of the polymeric nanostructures. In the context of vaccine construction, polymeric nanoparticles can be applied as controlled delivery systems of the immunomodulatory adjuvants and antigenic compounds. Antigens with an amphoteric nature can be attached by adsorption or surface immobilization to certain polymeric nanocarriers, such as chitosan or dextran sulfate-based nanoparticles. Release of the antigenic compound in such cases is predesigned based on the properties of the biological milieu (e.g., pH, temperature, ionic strength, etc.) [64,65].

Certain polymers, including dendrimers, polyamines, polyethylenimines, and copolymers, are functional materials capable of delivering mRNA and other vaccine candidates. Similar to other drug delivery systems, polymers can also protect encapsulated material from enzyme-mediated degradation and facilitate intracellular delivery. Nevertheless, the formulation of polymer-based vaccine nanoparticles tends to have high polydispersity [66]. To stabilize the formulation and improve the safety profile, structural modifications of polymer materials, such as incorporating hyperbranched groups, lipid chains, and biodegradable subunits, have been explored [67].

Despite the considerable potential of the polymeric nanocarriers and the magnitude of the bibliography on their therapeutic applications, a very limited number of polymeric constructs has reached clinical trials thus far [68].

### 4.2. Surfactant-Based Carriers

Non-ionic surfactant vesicles (NSVs), also known as niosomes, are relatively new drug delivery systems that are employed for managed and targeted delivery of drugs because of their ease of preparation on large scales and cost-effectiveness [69,70]. Moreover, their characteristics such as biocompatibility, biodegradability, and flexibility in structure have made them ideal carriers for loading hydrophobic and hydrophilic drugs [71]. These carriers have been applied in many areas, including gene transfer and vaccine delivery [72]. Despite all benefits of NSV delivery systems, their stability could be an issue with respect of the possibility of hydrolysis of the entrapped or encapsulated drug in an aqueous media. The problem of untimely and uncontrolled drug leakage from the non-ionic surfactant vesicles and aggregate formation of the niosomes may also occur. In order to alleviate these problems, incorporation of certain phospholipids and/or sterols (e.g., cholesterol—as a bilayer modulating agent) in the structure of niosomes is suggested [70,73,74,75].

### 4.3. Lipidic Delivery Systems

Lipid-based bioactive delivery systems are among the most promising technologies employed for the encapsulation and targeted release of drug and vaccine molecules. They include bilayer lipid vesicles (e.g., liposomes, nanoliposomes, archaeosomes, vesicular lipid gels, immunovesicles, lipospheres), solid lipid nanoparticles (SLN), tocosomes, and some other micro- and nanocarrier systems [76]. Figure 2 demonstrates the structure of available lipid-based drug and vaccine carriers.

#### 4.3.1. Liposomes as Drug and Vaccine Carriers

One of the first and most versatile drug delivery protocols is liposome, which is also known as a bilayer lipid or phospholipid vesicle. The word liposome refers to mesomorphic, nematic, and smectic structures that are mainly composed of lipid, phospholipid, and water molecules. The main ingredients of liposomes are amphiphilic lipid and phospholipid molecules [77]. They improve the stability and efficacy of bioactive compounds by entrapment and release of water-soluble, lipid-soluble, and amphiphilic materials, as well as targeting the encapsulated compounds to particular cells or tissues inside the body. Liposomes can be made on a small scale (e.g., for lab-scale research and development), pilot-scale, or industrial scales using natural ingredients, such as egg or soy lecithin. It is also possible to include other ingredients, such as sterols (cholesterol or plant-derived phytosterols), polypeptides (e.g., antigens), antioxidants (e.g., α-tocopherol), as well as polymers (such as poly-ethylene-glycol or chitosan) in the structure of the lipid vesicles. Figure 3 depicts main components used in the construction of a liposomal or nanoliposomal vaccine formulation. These additives assist in modulating the bilayer structure of liposomes, increase their tolerance towards reactive oxygen assaults, increase their blood circulation half-life, and provide a targeting strategy for the lipid vesicles [78]. Different types of liposomal vaccine uptake mechanisms are illustrated in Figure 4.

#### 4.3.2. Nanoliposomes

Nanoliposomes (nanometric versions of liposomes or lipidic nanovesicles) can be briefly defined as colloidal nanostructures composed of lipid or phospholipid molecules [79]. It should be noted that liposomes and their nano-versions possess the same physical, chemical, and thermodynamic properties, which are mainly governed by their ingredients and their suspension media. The difference between the two related drug carriers comes from their size range. It is known that the smaller the particle size, the larger the surface-to-volume ratio they will acquire. Therefore, in comparison with liposomes, nanoliposomes provide more surface area and have more potential to increase solubility, enhance bioavailability, improve the controlled release of the loaded material, and enable accurate targeting of the encapsulated drug or vaccine moieties. Manufacture of both liposomes and nanoliposomes involves input of energy (in the form of agitation, heat, ultrasound, etc.) to a dispersion of lipid and phospholipid molecules in an aqueous medium [77,78,79,80,81]. Although liposomes and nanoliposomes are initially prepared as a liquid product (aqueous suspension), they can be subsequently incorporated in a cream, lotion, aerosol, soft-gel, powder (via lyophilization or spray drying for instance), or other formulations and dosage forms [76]. These versatilities open significant opportunities for efficient vaccine design and formulation.

#### 4.3.3. Solid Lipid Nanoparticles (SLNs)

Solid lipid nanoparticles can be defined as nano-sized colloidal carriers made up of accurate ratios of lipid, surfactants, and bioactive compounds. These relatively new drug delivery systems are superior to other available drug carriers, due to their small size and lipid core that provide some unbeatable properties. They also provide a suitable alternative to the existing drug delivery technologies, such as liposomes, emulsions, and polymeric nanocarriers. In these new generations of submicron-sized lipidic carriers, the liquid-phase lipid has been replaced by a solid-phase lipid, resulting in the lipidic particle that is solid at room temperature or body temperature [80]. Triglycerides, acetyl alcohol, emulsifying wax, beeswax, carnauba wax, cholesterol, and cholesterol butyrate are some of the solid lipids employed in the preparation of solid lipid nanoparticles [76,81].

#### 4.3.4. Immunostimulatory Complexes (ISCOMs)

The immune stimulatory complexes, known as ISCOMs, are other vaccine delivery carriers with potent adjuvanticity, as attested clinically [82]. They size about 40–60 nm and are self-assembled in solution at well-defined ratios of saponin adjuvant Quil A and a protein antigen, cholesterol, and phospholipid. The neutral or positively charged dimethylaminoethane-carbamoyl (DC)-cholesterol, and zwitterionic phospholipids or the cationic dioleoyl-trimethyl-ammonium-propane (DOTAP) are such materials to be used for ISCOM formation. These cage-like particles with a hollow center are also used for trapping hydrophobic antigens [82,83].

ISCOMs-based vaccines possess the potentiality to boost both humoral and cellular immune responses. ISCOMATRIX is a vaccine delivery carrier and adjuvant, which does not contain antigen and operates similarly to ISCOMs. Nevertheless, it is more applicable than ISCOMs due to its potency to remove the limitation of hydrophobic antigens. Antigens added later to the preformed ISCOMATRIX particles can be employed to entrap hydrophilic antigens [84,85]. Several research groups have published the usage of different antigens for forming ISCOMs and ISCOMATRIX vaccines, including antigens derived from HIV [84,85], HPV, Newcastle disease, and influenza [86,87].

### 4.4. Tocosome as a Potential Vaccine Carrier

Tocosome is a vesicular and colloidal bioactive carrier system that is mainly composed of phosphate-group-bearing alpha tocopherols [88]. Tocosomes can also accommodate proteins, polymers, and sterols in their structures, as is the case with liposomes and nanoliposomes. Alpha-tocopherol phosphate (TP), a main ingredient of tocosome, is the phosphorylated form of alpha-tocopherol that is available naturally in human and some animal tissues and in certain food compounds (such as fruits, green vegetables, cereals, dairy products, as well as deferent nuts and seeds) [89,90]. TP is made up of a phosphate group binded to one hydrophobic chain (phytyl tail), composed of three isoprene units. Di-alpha-tocopherol phosphate (T_2_P), a molecule related to TP, is composed of two phytyl chains. The hydrophobic phytyl chains of T2P molecules cannot arrange in a parallel position because of the presence of bulky isoprene side-chains (unlike phosphatidylcholine and some other phospholipids). Accordingly, T_2_P possesses a conical geometric shape, while the TP has a cylindrical geometric shape, similar to PC [91]. Clinical studies have demonstrated that TP and T_2_P molecules have many advantages for human health, including atherosclerotic prevention as well as cardioprotective and anti-inflammatory properties [92,93]. In addition, the TP molecule has an inhibitory effect against tumor invasion [94]. Investigations have also reported that TP can protect primary cortical neuronal cells from glutamate-induced cytotoxicity in vitro and can decrease the amount of lipid peroxidation products in the plasma and liver of mice in vivo [95]. Formulations of tocosomes include various phospholipid molecules (in addition to TP and T_2_P components) and different combinations of cholesterol. They have recently been used for the entrapment and controlled release of the anticancer drug 5-fluorouracil successfully [88].

## 5. Lipidic Carrier-Based Vaccine Design and Formulation

Lipids, lipid-like compounds, tocopheryl-phosphates, and lipid derivatives have been widely used to formulate nanovesicles and nanoparticles for in vivo delivery of vaccines [96,97,98,99,100]. Lipidic nanocarriers are generally defined as nano-sized particulate or colloidal systems that are composed of natural, synthetic, or physiological lipid/phospholipid materials [101]. Lipidic nanocarriers are being developed for vaccine delivery for the following two main reasons. Firstly, they can encapsulate vaccine molecules and compounds, and hence, protect them from enzymatic degradation [96]; in the case of mRNA vaccines, the reported encapsulation efficiency by lipidic nanocarriers was usually high, designating the mRNA molecules were mostly internalized within the vesicles [102,103]. Secondly, the nanocarriers can effectively transport vaccine molecules into the cell cytosol through a series of endocytosis processes [104]; for example, it was reported that the surface adsorption of apolipoprotein E on the lipid-based nanoparticles might facilitate its intracellular delivery via low-density lipoprotein, receptor-mediated, clathrin-dependent endocytosis [105]. This endocytosis process delivered the vaccine-loaded nanocarriers to the cell membrane-bound vesicles, including endosomes and lysosomes [104,106]. Eventually, the lipid-based nanocarriers facilitated translocation of the vaccine cargos into the cytosol for protein expression [97].

The lipidic nanocarriers usually contain one or more of the functional lipid components, which are crucial for the intracellular vaccine delivery [66,97,107]. The cationic or ionizable lipid materials, such as 1,2-di-O-octadecenyl-3-trimethylammonium propane (DOTMA), *N*,*N*-Dimethyl-2,3-bis[(9Z,12Z)-octadeca-9,12-dienyloxy]propan-1-amine (DLinDMA), and *N*1,*N*3,*N*5-tris(3-(didodecylamino)propyl)benzene-1,3,5-tricarboxamide (TT3) usually contain one or more amino groups [108,109,110,111]. These lipid molecules can be positively charged at a certain pH to encapsulate the negatively charged molecules (e.g., DNA or RNA) via electrostatic interactions and help interact with the cell membrane on the target cells. Recent studies indicated that the final step of vaccine release from lipidic nanocarriers into the cytosol might involve the membrane disruption of endosomes. In this process, the ionizable cationic lipids were suggested to interact with anionic phospholipids on the endosomal membrane and form disruptive non-bilayer structures that finally released the encapsulated vaccine into the cell cytoplasm [112]. Moreover, the structure–activity relationship of the lipidic structures for vaccine delivery and endosomal escape was studied. Results of these studies indicated that the hydrophilic head group of the phospholipid molecules might determine the acid dissociation constant (pKa) and influence the vaccine delivery efficiency [113]. Besides, modification of fatty acids structures in the hydrophobic tails may also affect the vaccine delivery efficiency [113,114]. Even though the membrane disruptive features of the lipid molecules improve the delivery efficiency, certain synthetic materials may cause side effects in vivo [115]. The helper lipids, such as 1,2-dioleoyl-sn-glycero-3-phosphoethanolamine (DOPE), 1,2-distearoyl-sn-glycero-3-phosphocholine (DSPC), and cholesterol, stabilize the structures of lipidic nanocarriers and facilitate endosomal escape [116]. The PEG-lipid conjugates could stabilize the nanocarriers during preparation and provide a hydrophilic outer layer that prolongs the circulation time after in vivo administration [66,116,117,118]. In addition to these functions, the engineered ionizable lipid molecules containing cyclic amino head groups, isocyanide linker, and two unsaturated alkyl tails were reported to provide adjuvant activities independent of the encapsulated vaccine. These cyclic amino head groups directly bound the STING (stimulator of interferon genes) protein and triggered the downstream signaling pathway, leading to an elevated innate response. After subcutaneous injection of an antigen-coding mRNA encapsulated by such lipidic nanocarriers into mice, the researchers observed the upregulation of antigen-specific T cells and inhibition of tumor growth [119].

The formulation methods of lipid-based vaccines mainly include thin-film hydration [120,121], direct mixing [122], ethanol injection, and continuous-flow microfluidic device [123,124]. Among these methods, the continuous-flow microfluidic device emerges as a prevalent method to prepare RNA encapsulated nanoparticle, especially lipid nanoparticles, for in vivo applications [66,125]. These chip-based microfluidic devices mix two laminar flows, the vaccine-containing aqueous phase, and the carriers-containing solvent phase, through a confined microchannel equipped with mixers at a controlled speed, leading to rapid diffusion, change of polarity, and self-assembly of vaccine-lipidic nanoparticles at the interface [126,127]. The resulting lipid nanoparticles are relatively homogeneous formulation and usually show spherical and multilamellar morphologies [127]. Compared to other preparation methods, the use of continuous-flow microfluidic devices increases reproducibility, improves molecular stability, reduces the chance of contamination, and is easily scaled up for preclinical and clinical studies [125,128]. Another safe, mild, and robust method of vaccine preparation on industrial scales is “Mozafari method” [129], which is based on a former technique known as heating method [130,131]. This patented process, and its modifications, does not require application of toxic solvents, detergents, high-shear forces, or extreme pH values [132,133]. Drug and vaccine formulations can be mass manufactured using a simple vessel at very low shear forces (e.g., maximum agitation power of 1000 rpm) and temperatures below 40 °C [134,135,136].

## 6. Vaccine Targeting Strategies

Covering the surface of lipidic nanocarriers with immune cell receptors may facilitate their uptake by the desired type of immune cells. For the immune system to be activated, or for an immune response to be provoked, specific APCs need to encounter an antigen and a risk signal. APCs are concentrated at high concentration in the lymph nodes. For lymph node targeting, the vaccine moiety can be injected directly in the nodes, or the nanocarriers can be designed to accumulate in the lymph nodes. The two most important parameters for lymph node accumulation are size and surface composition of the nanocarriers. Scientific reports indicate decreasing lymphatic uptake with increasing particle size. Only nanoparticles with diameters smaller than about 150 nm appear to enter the lymphatic capillaries, and are subsequently drained to the peripheral lymphatics [137,138,139]. On the other hand, larger lipid particles and vesicles are retained at the injection site [140,141]. Larger vesicles are believed to be recognized and cleared more rapidly by the complement system because they present a larger number of recognition sites on their surface [142]. Intravenously injected lipidic nanoparticles are mostly eliminated from the blood circulation, after opsonization, by the cells of the immune system, which are mainly present in the liver and spleen. Opsonization is the process by which particles are labeled for ingestion and elimination by the macrophages and phagocytes. Inclusion of certain polymers (e.g., polyethylene glycol (PEG), polyhydroxyethyl L-asparagine (PHEA)), or incorporation of glycolipids (e.g., monosialoganglioside) to the lipid vesicles, prevent opsonization and result in sterically stabilized vesicles. These carrier systems have several advantages over conventional formulations including reduced recognition and uptake by MPS/RES, extended circulation half-lives, and dose-independent pharmacokinetics (for a recent review see: [143]). It has been reported that anti-PEG antibody response following repeated intravenous (IV) administration of PEGylated vesicles dramatically accelerate blood clearance of the vesicles and leads to acute hypersensitivity [144,145]. This finding is a concern for immunotherapy applications, where multiple dosing may be required for long-lasting protection. A possible solution may be achieved by modifying the PEG molecule into a less immunogenic variant, using alternative molecules, such as PHEA, or by using different administration routes.

Another group of target cells for vaccination purposes are dendritic cells (DC). These cells are covered with different receptors, which include lectins that recognize carbohydrate moieties present on many pathogens, and are involved in antigen capture and presentation [146,147]. Other dendritic cell receptors are the mannose receptor, DC-SIGN, DEC-205, and Langerin [148,149,150,151]. Active targeting of dendritic cells has been extensively studied in recent years. The term active targeting is somewhat misleading in that the lipidic nanoparticles are not actually actively guided toward dendritic cells. In fact, uptake by dendritic cells is enhanced by modification of the surfaces of vesicles with suitable molecules. These molecules include antibodies (anti-CD11c, anti-DEC-205) or peptides (P-D2, RGD), where anti-CD11c antibody, P-D2, and RGD peptides target integrins and anti-DEC-205 antibody targets the c-type lectin receptor DEC-205 [152].

DC receptors have been characterized and utilized for targeted protein and protein–lipid-nanoparticle vaccines extensively. Initial studies employed mannose monosaccharides or disaccharides to target vaccines to dendritic cells, albeit with little success [153]. The binding affinity of such monosaccharaide molecules is very weak, typically in the mM range. However, the binding affinity can be enhanced by orders of magnitude by coupling the monosaccharaides to a scaffold that forms a multivalent cluster or by using multibranched saccharides [154,155]. Biodistribution and pharmacokinetics of such engineered nanoparticles is altered significantly when varying the density of the sugar moieties [155]. A high dendritic cell specificity was observed for LNPs containing 11% mannosylated lipids, while no specificity was observed for LNPs containing 3% mannosylated lipids [156]. In another study, mannosylated mRNA–lipid nanoparticles coding for MART-1 also showed higher vaccination rates compared with their nonmannosylated analogs [157]. It would be interesting to investigate if surface modifications with ligands for different DC subsets also increase the potency of liposomal mRNA vaccines, as has been shown in the case of peptide-based vaccines [158].

## 7. Synopsis

Recent progress in nanobiotechnology and advanced nanomanufacturing lay the foundation for rapid development of innovative vaccine technologies to make an impact during the COVID-19 pandemic. Towards this end, efficient vaccine encapsulation and delivery systems with the capability of mass manufacturing the vaccine formulation play a critical role. However, improvements are still required to optimize the safety profile and to increase the vaccination efficacy. The progress in the development of various delivery systems has enabled numerous preclinical studies and clinical trials. Among the all-available drug delivery systems, lipidic carriers represent one of the most advanced platforms for vaccine delivery and targeting in vivo. Compared with other delivery protocols, the lipidic carriers have the longest history of being under research and development worldwide as well as having the highest number of FDA approved pharmaceutical products on the market. As the delivery methods and the vaccine formulations further advance, theranostic lipidic vaccines will become an important class of medicine to effectively tackle diverse health issues, such as viral and other infectious diseases.

## Figures and Tables

**Figure 1 biomedicines-09-00520-f001:**
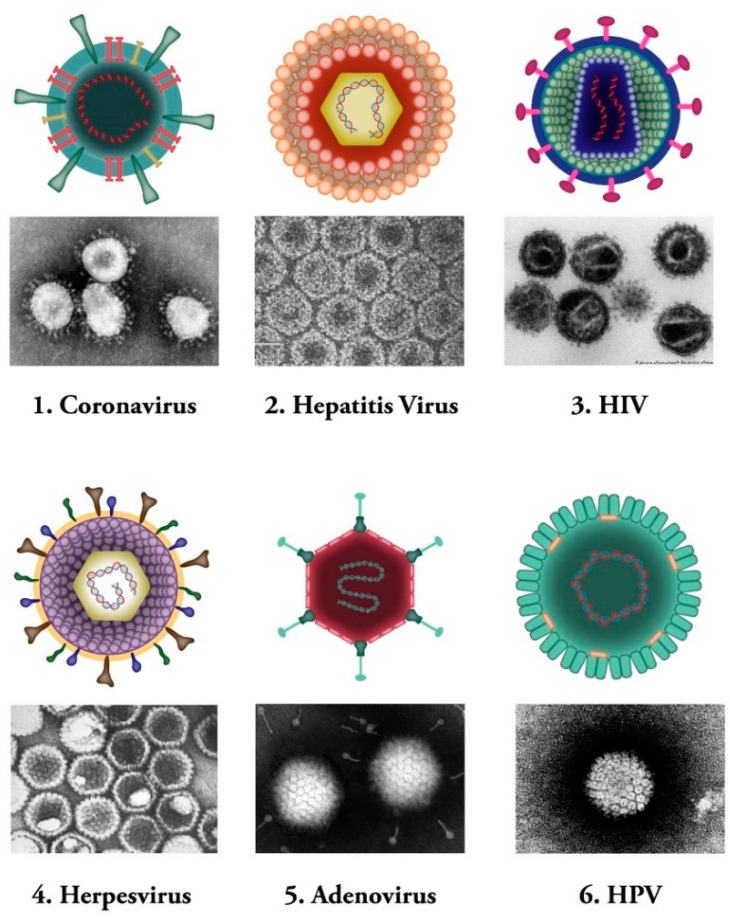
Common human viral pathogens. Six types of human infecting viruses, including coronavirus, hepatitis virus, human immunodeficiency virus (HIV), herpesvirus, adenovirus, human papilloma virus (HPV) are depicted schematically or as observed under electron microscopes.

**Figure 2 biomedicines-09-00520-f002:**
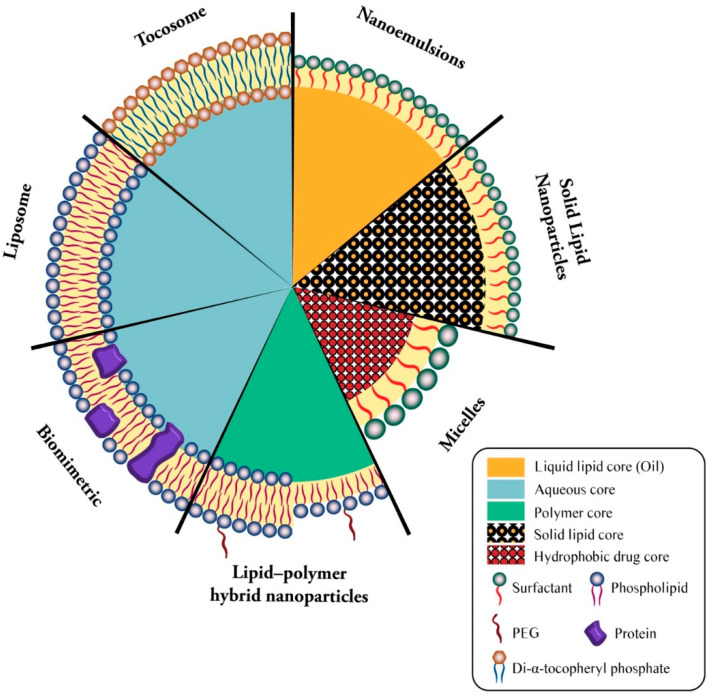
Simplified structure of main lipidic carrier systems. Seven types of lipidic vaccine/drug carriers consists of liposome (phospholipid bilayer + aqueous core), biomimetric (proteinated phospholipid bilayer + aqueous core), lipid-polymer hybrid nanoparticles (bi- or monolayer phospholipid + polymer core), micelles (surfactant monolayer + hydrophobic drug core), solid lipid nanoparticles (surfactant monolayer + solid lipid core), nanoemulsion (surfactant monolayer + liquid lipid core), and tocosome (Di-α-tocopheryl phosphate bilayer + aqueous core).

**Figure 3 biomedicines-09-00520-f003:**
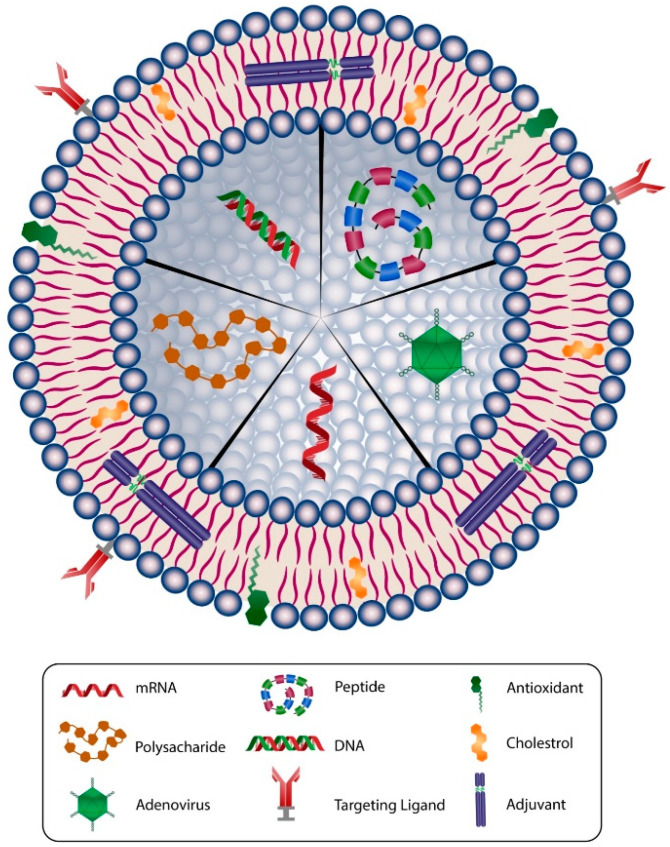
Schematic representation of main components used in a typical liposomal/nanoliposomal vaccine formulation. Liposomes can carry different types of drug or vaccine core, such as mRNA, DNA, peptide, adenovirus, or polysacharide. The membranes of liposomal carriers are made of phospholipids, cholesterol, targeting ligands, antioxidants, and adjuvants.

**Figure 4 biomedicines-09-00520-f004:**
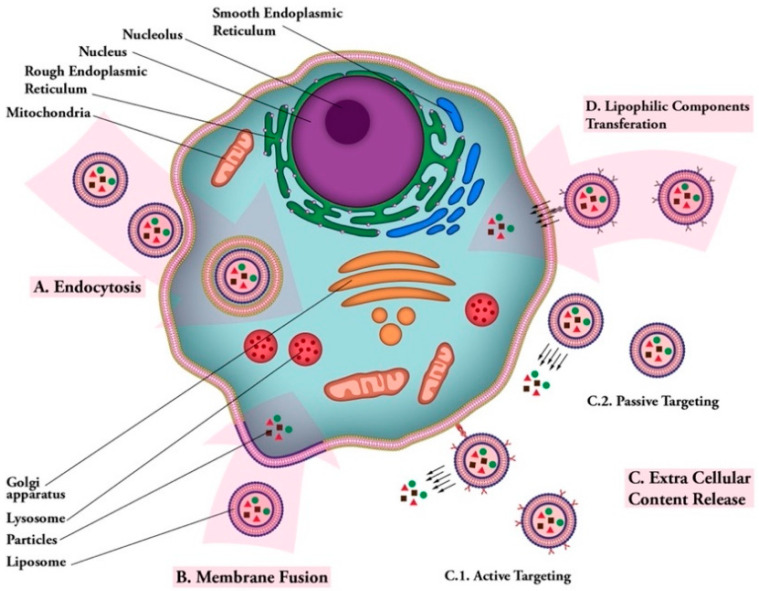
Liposomal vaccine uptake mechanisms. (**A**) Endocytosis: uptake of vaccine carrier by cell; (**B**) membrane fusion: fusion of vaccine carrier and cell membranes and release of vaccine particles into cytoplasm; (**C**) extra cellular content release: (**C.1**); active targeting: releasing vaccine particles near cell membrane after targeting via ligand-receptor connection. (**C.2**); passive targeting: release of vaccine particles near cell membrane; (**D**) lipophilic components transferation: transfer of lipophilic vaccine particles through cell membrane after carrier ligand-receptor binding.

**Table 1 biomedicines-09-00520-t001:** Different types of currently available protocols employed in vaccine construction.

Vaccine Type		Target Pathogen/Disease
Nucleic acid based vaccine		SARS-Cov-2
Peptide based vaccine		SARS-Cov-2, Hepatitis B
Pathogen based vaccine	Live attenuated viral vaccine	Vaccina (smallpox), Measles, mumps, and rubella (MMR combined vaccine), Varicella (chickenpox), Influenza (nasal spray), Rotavirus, Zoster (shingles), Yellow fever
	Inactivated viral vaccine	Polio (IPV), Hepatitis A, Rabies
	Viral subunit vaccine	Hepatitis B, Influenza (injection), Haemophilus influenzae type b (Hib), Pertussis (part of DTaP combined immunization), Pneumococcal, Meningococcal, Human papillomavirus (HPV)
Adjuvant		Human papillomavirus (HPV) types 16 and 18, influenza (flu), Hepatitis B

## Data Availability

Not applicable. This Review Article analyzed and discussed already published data and did not report any new data.

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
