# Peer review of "Methodical Design of Viral Vaccines Based on Avant-Garde Nanocarriers: A Multi-Domain Narrative Review"

_biomedicines, 2021, doi:10.3390/biomedicines9050520_

Round 1

Reviewer 1 Report

A well-prepared manuscript. Careful and detailed drawings. Lots of interesting and meaningful references. I have no major objections to this article. 

Author Response

We thank this Reviewer for his highly encouraging statement:

“A well-prepared manuscript. Careful and detailed drawings. Lots of interesting and meaningful references. I have no major objections to this article.”

Reviewer 2 Report

I appreciate the opportunity to serve as a reviewer for the manuscript titled “Methodical design of viral vaccines based on theranostics nanocarriers: a multi-domain narrative review”. Given the current pandemic situation, it is an interesting topic to discuss vaccine delivery to human.

The manuscript presents a good overview of fundamentals of virology and vaccination. The graphical figures presented in the manuscript are vivid and neat. However, I do believe that the manuscript can benefit from additional modifications. Please see below for my recommendations

  1. The manuscript doesn't include discussion about theranostics, which is a culmination of therapeutic and diagnostic substance in the same carrier. To that effect, I strongly recommend the authors to consider removing the word theranostic from the title. In general, the manuscript extensively discusses lipid nanocarriers, it will be prudent to consider adding that to the title.
  2. In the context of vaccination, referencing table 1 on page 2 appears to be pre-mature. I recommend the authors to reference it in section 2 where different types of vaccines are discussed.
  3. "Nanovesicles are able to function as adjuvants for these vaccines", I suggest the authors to substantiate this statement with appropriate references.
  4. Sections 1,2 and 5,6 are well written however sections 3 and 4 require some modification in the way the discussion is presented. Except for the lipid carriers, there is less weightage given to the discussion of other nanocarriers.
  5. Section 4.1, para 2 "Similar to functional lipid-based carriers, polymers can also protect encapsulated material from enzyme-mediated degradation and facilitate intracellular delivery." comparison with lipid based carriers is premature since lipid based carrier is mentioned in subsequent sections. I suggest rewording this statement.
  6. Section 4.1 needs more details. The authors have not listed or reviewed in detail the type of polymers that are used for vaccine delivery.
  7. Liposomal system review in two sections 4.3.1 and 4.3.2 can be combined under single heading, since "liposome" in the broad context includes both nano-and micron sizes.   
  8. Section 4.3.4: ISCOMs, the authors have not sufficiently described what this carrier system is made of? it is classified under lipidic carriers, however additional details about it's material make-up is required.  
  9. The authors have repeatedly mentioned that nanocarriers have several advantages, but they have not discussed what are those advantages in any part of the manuscript. 
  10. It will also benefit the readers if the authors will comment on the physical/chemical stability of each of these systems. 

Author Response

We thank this Reviewer for his instructive comments and suggestions. Please find our answers to each comment below:
1.    The manuscript doesn't include discussion about theranostics, which is a culmination of therapeutic and diagnostic substance in the same carrier. To that effect, I strongly recommend the authors to consider removing the word theranostic from the title. In general, the manuscript extensively discusses lipid nanocarriers, it will be prudent to consider adding that to the title.
We agree with the Reviewer and have removed the word “theranostic” from the title and replaced it by “avant-grade” due to the fact that the manuscript also reviews non-lipidic contemporary, modern vaccine carrier systems. 

2.    In the context of vaccination, referencing table 1 on page 2 appears to be pre-mature. I recommend the authors to reference it in section 2 where different types of vaccines are discussed.
We comply with the suggestion and have moved Table 1 to section 2.

3.    "Nanovesicles are able to function as adjuvants for these vaccines", I suggest the authors to substantiate this statement with appropriate references.
Two References are now added to the statement as suggested. 

4.    Sections 1,2 and 5,6 are well written however sections 3 and 4 require some modification in the way the discussion is presented. Except for the lipid carriers, there is less weightage given to the discussion of other nanocarriers.
Substantial improvements to the mentioned sections have now been performed as per Reviewer’s suggestions. 

5.    Section 4.1, para 2 "Similar to functional lipid-based carriers, polymers can also protect encapsulated material from enzyme-mediated degradation and facilitate intracellular delivery." comparison with lipid based carriers is premature since lipid based carrier is mentioned in subsequent sections. I suggest rewording this statement.
The mentioned sentence is now amended. 

6.    Section 4.1 needs more details. The authors have not listed or reviewed in detail the type of polymers that are used for vaccine delivery.
Examples of polymers used are now expanded to:
Chitosan, dextran sulfate-based nanoparticles, dendrimers, polyamines, polyethylenimines and copolymers. 

7.    Liposomal system review in two sections 4.3.1 and 4.3.2 can be combined under single heading, since "liposome" in the broad context includes both nano-and micron sizes.   
Authors of the manuscript, being pioneers in the field of liposomology (publications date back to 1994) and being authors of the first book on “nanoliposomes” (Nanoliposomes: from fundamentals to recent developments. Trafford Publication, Oxford, 2005) strongly believe that the terminology “liposome” should not be used randomly in place of the word “nanoliposome” and each terminology should be used in a scientific manner to distinguish between micrometre-sized vesicles and nanometric vesicles. Consequently, we strongly believe that the mentioned 2 sections should not be merged. 

8.    Section 4.3.4: ISCOMs, the authors have not sufficiently described what this carrier system is made of? it is classified under lipidic carriers, however additional details about it's material make-up is required.  
More details are added to the section as suggested by the Reviewer.

9.    The authors have repeatedly mentioned that nanocarriers have several advantages, but they have not discussed what are those advantages in any part of the manuscript. 
More explanations are now provided in the manuscript for the advantages of nanocarriers as suggested. Some examples include:

Nanoparticles and nanovesicles, due to their intrinsic adjuvanticity (by activating complement system, inducing autophagy and activation of inflammasome) are also considered as vaccine adjuvants nanosystems …

A broad range of active moieties can be encapsulated and delivered by nanovesicles and nanocarriers. These nanosystems possess more surface-area to volume ratio when com-pared to micrometric drug carriers and as such they can attach to / interact with the target cells more efficiently. Nanotechnology-based vaccines can be formulated employing nucleic acids, purified subunit proteins or polysaccharides, synthetic peptides, or recombinant proteins.

Nanotechnology and nanomedicine provide substantial advantages for the formulation of vaccine protocols and enable either the targeted delivery or co-delivery of the therapeutic molecules and reinforce the delivery of high-value and high-tech bio-therapeutic agents such as proteins or genetic material …

10.    It will also benefit the readers if the authors will comment on the physical/chemical stability of each of these systems.
The issue of stability has been discussed in several occasions in the manuscript, including:

Under Section 3:
They can also increase immunogenicity, cellular uptake and stability of vaccines by encapsulation or absorption of the vaccine antigen or DNA in a proper formulation.

Under Section 4.2:
Despite all benefits of NSV delivery systems, their stability could be an issue with respect of the possibility of hydrolysis of the entrapped or encapsulated drug in an aqueous media.

Under Section 4.3.1:
They improve the stability and efficacy of bioactive compounds by entrapment and release of water-soluble, lipid-soluble and amphiphilic materials, as well as targeting the encapsulated compounds to particular cells or tissues inside the body.

Under Section 5:
Compared to other preparation methods, the use of continuous-flow microfluidic devices increases reproducibility, improves molecular stability, reduces the chance of contamination, and is easily scaled up for preclinical and clinical studies.

Reviewer 3 Report

Congratulations to the authors for putting together a comprehensive introduction and detailed descriptions of nanocarriers.

Please modify the manuscript as below:

Comments on 2.1. Nucleic acid-based strategies:

The sentence “The synthetic antigens stimulate the protein production” is misleading, it could be “synthetic DNA/RNA stimulates the protein production”.

The sentence “This protein locates in the plasma membrane and protein modification processes can take place precisely” needs to be rephrased to convey the message.

Comments on 2.3.1.4. Vaccine Delivery Techniques:

The sentence “in addition to a clear understanding of the method of action of the delivery techniques are also desirable factors” needs to be rephrased.

Author Response

We thank this Reviewer for his instructive comments and suggestions and encouraging statement:
“Congratulations to the authors for putting together a comprehensive introduction and detailed descriptions of nanocarriers.”

Please find our answers to each comment below:

1.    Comments on 2.1. Nucleic acid-based strategies:
The sentence “The synthetic antigens stimulate the protein production” is misleading, it could be “synthetic DNA/RNA stimulates the protein production”.
We agree with the Reviewer and have amended the sentence to:
“The synthetic DNA / RNA stimulate the protein production, similar to the infection situation, in the target cells.”

2.    The sentence “This protein locates in the plasma membrane and protein modification processes can take place precisely” needs to be rephrased to convey the message.
As suggested by the Reviewer, the sentence has been changed to:
“The synthesized protein will be placed in the cytosolic plasma membrane (e.g. endo-plasmic reticulum) and protein modification processes can precisely take place [26].”

3.    Comments on 2.3.1.4. Vaccine Delivery Techniques:
The sentence “in addition to a clear understanding of the method of action of the delivery techniques are also desirable factors” needs to be rephrased.
As suggested by the Reviewer, the sentence has been changed to:
“Acceptable bio-compatibility, biodegradability, and safety aspects of the formulation, in addition to a clear understanding of the mechanism of action of the vaccine delivery system, are also desirable factors.”

Round 2

Reviewer 2 Report

Rebutal acceptable.